# Transcriptome Profiling of Different Developmental Stages on Longissimus Dorsi to Identify Genes Underlying Intramuscular Fat Content in Wannanhua Pigs

**DOI:** 10.3390/genes14040903

**Published:** 2023-04-12

**Authors:** Xiaojin Li, Yanan Yang, Lei Li, Man Ren, Mei Zhou, Shenghe Li

**Affiliations:** 1College of Animal Science, Anhui Science and Technology University, Chuzhou 233100, China; 2Anhui Province Key Laboratory of Animal Nutritional Regulation and Health, Chuzhou 233100, China; 3Institute of Animal Husbandry and Veterinary Medicine, Anhui Academy of Agricultural Sciences, Hefei 230041, China

**Keywords:** Wannanhua pig, IMF, transcriptome analysis, longissimus dorsi muscle, RT-qPCR

## Abstract

Intramuscular fat (IMF) is a key index to measure the tenderness and flavor of pork. Wannanhua pig, a famous indigenous pig breed in Anhui Province, is renowned for its high lipid deposition and high genetic divergence, making it an ideal model for investigating the lipid position trait mechanisms in pigs. However, the regulatory mechanisms of lipid deposition and development in pigs remain unclear. Furthermore, the temporal differences in gene regulation are based on muscle growth and IMF deposition. The purpose of this study was to study the expression changes of longissimus dorsi (LD) at different growth stages of WH pigs at the molecular level, to screen the candidate genes and signaling pathways related to IMF during development by transcriptome sequencing technology, and to explore the transcriptional regulation mechanism of IMF deposition-related genes at different development stages. In total, 616, 485, and 1487 genes were differentially expressed between LD60 and LD120, LD120 and LD240, and LD60 and LD240, respectively. Numerous differentially expressed genes (DEGs) associated with lipid metabolism and muscle development were identified, and most of them were involved in IMF deposition and were significantly up-regulated in LD120 and LD240 compared to LD60. STEM (Short Time-series Expression Miner) analysis indicated significant variations in the mRNA expression across distinct muscle development stages. The differential expression of 12 selected DEGs was confirmed by RT-qPCR. The results of this study contribute to our understanding of the molecular mechanism of IMF deposition and provide a new way to accelerate the genetic improvement of pork quality.

## 1. Introduction

Fat is the main tissue for energy storage in animals, and it is mainly involved in important physiological processes such as lipid metabolism, system energy homeostasis, and systemic insulin sensitivity [1]. Intramuscular fat (IMF) refers to fat in the skeletal muscle and is composed of fat droplets within muscle cells and adipose tissue between muscle fibers [2]. IMF in the LD muscle is important for determining pork quality, affecting muscle tenderness, flavor, and juiciness [3]. Appropriate IMF can produce a better taste and is important for determining the meat quality. IMF deposition primarily occurs through the increase in the numbers of preadipocytes (cell proliferation) and adipocyte size (cell enlargement) [4]. Intramuscular preadipocytes proliferate and differentiate into mature adipocytes, after which the number of intracellular lipid droplets and the cell volume increases, forming fat deposits [5]. The IMF increases gradually with the increase in age [6]. Therefore, identifying the genetic mechanism regulating intramuscular adipogenesis during pig growth and development is crucial to better understand pig fat deposition.

Compared with western pig breeds, Chinese indigenous pigs have a slow growth rate and low lean meat rate but have excellent meat quality and superior lipid deposition. The population of Chinese indigenous pig breeds is small, the breeding method is relatively closed, and the selection intensity is low, resulting in obvious genetic and phenotypic differentiation [7]. The Wannanhua (WH) pig is an excellent indigenous pig breed from Anhui Province. The marbling, flesh color, and IMF content of WH pigs are significantly higher than those of western pig breeds (*p* < 0.01) [8,9]. Therefore, the WH pig is an ideal model for studying lipid deposition. The developmental process of porcine skeletal muscle is divided according to time, including the embryonic muscle fiber generation stage and postnatal muscle fiber type transformation and maturation stage. After birth, the muscle fiber type changes, with the diameter thickening and elongation mainly occurring. The main time period for the transformation of the muscle fiber type is from 0 to 60 days after birth. Most of the newborn piglets’ muscle fibers are oxidized, and the proportion of oxidized muscle fibers decreases gradually during 1–4 weeks after birth, while the proportion of glycolic muscle fibers increases sharply [10,11]. The IMF content in pigs increases with age. Therefore, three different developmental stages of WH pigs at 60, 120, and 240 days of age were selected as the research subjects in this study.

The study of pork quality and IMF using transcriptomics technology has made great progress in recent years. Zhao et al. [12] performed a skeletal muscle tissue analysis of Tongcheng and Yorkshire pigs during 11 developmental stages. However, previous transcriptomic analyses of porcine skeletal muscle tissue have focused on identifying specific genes affecting the muscle development across two different breeds, and the differences in the dynamic transcriptome profiles of the LD muscle at different developmental stages in WH pigs have been rarely studied.

In this study, high-throughput transcriptome sequencing was used to study the LD muscle tissue of WH pigs at different developmental stages to identify candidate genes and key pathways for lipid deposition, and to further study lipid regulatory networks that could possibly drive postnatal fat deposition in pigs. The molecular mechanisms investigated promote the developmental processes of Chinese indigenous pig breeds, especially WH pigs, which will help to improve the pork quality in future breeding.

## 2. Materials and Methods

### 2.1. Ethics Statement and Collection of Tissue Samples

Experiments were performed according to the Regulations for the Administration of Affairs Concerning Experimental Animals and approved by the Animal Research Committee of Anhui Science and Technology University. A total of nine castrated male WH pigs were randomly selected from the Huangshan Huizhou Youyou Pig Agriculture Development Co. Ltd. (Huangshan, Anhui, China) at 60, 120, and 240 days to cover three developmental stages (*n* = 3). Pigs were reared under the same conditions. Three individuals of similar body weight were used for each stage. Commercial feed and water were offered ad libitum. The pigs were housed in pens with concrete flooring, nipple bowl drinkers, and feeders. A section of the LD muscle close to the third or fourth lowest rib was dissected from each pig immediately after slaughter. Parts of the samples (0.3–0.4 cm thick) were immediately frozen in liquid nitrogen, transferred to the laboratory, and stored in a −80 °C refrigerator for later RNA-seq, and part of the samples were placed in an incubator in an ice pack and then quickly sent to the laboratory for IMF content determination.

### 2.2. RNA Extraction, Library Construction, and Illumina Sequencing

RNA libraries were constructed and deep-sequenced by the Shanghai Biotechnology Corporation (Shanghai, China). The total RNA was extracted using MagBeads Total RNA Extraction Kit (Cat# T02-096, Life technologies, Carlsbad, CA, USA) and checked for an RIN number to inspect the RNA integrity by an Agilent 2100 Bioanalyzer (Agilent technologies, Santa Clara, CA, USA). The qualified total RNA was further purified by an RNA Clean XP Kit (Cat# A63987, Beckman Coulter, Inc. Kraemer Boulevard, Brea, CA, USA) and RNase-Free DNase Set (Cat# 79254, QIAGEN, GmBH, Frankfurt, Germany). Purified mRNA was first fragmented by the RNA fragmentation kit (Ambion, Thermo Fisher Scientific Inc., Waltham, MA, USA). The mRNA fragments were then used as templates to synthesize the first-strand cDNA with a random hexamer primer and reverse transcriptase (Invitrogen, Carlsbad, CA, USA). Synthesis of the second-strand cDNA was catalyzed by RNase H and DNA polymerase Ⅰ. The cDNA fragments were purified with a QiaQuick PCR extraction kit. Sequencing adaptors were ligated to the purified fragments, which were then washed with EB buffer, to achieve end repair and single-nucleotide A (adenine) addition. The resultant 200–700-bp-long fragments were purified by agarose gel electrophoresis and selected as sequencing templates for PCR amplification. Finally, the sequenced reads from the nine libraries (LD60-1, LD60-2, LD60-3 and LD120-1, LD120-2, LD120-3 and LD240-1, LD240-2, LD240-3) were analyzed using an Illumina HiSeqTM 2000 platform.

Low quality reads were removed by Perl script, and the clean reads were filtered from the raw reads and mapped to the Sus scrofa genome (Sscrofa11.1) using HISTA2 (version:2.0.4) software (11 November 2021) [13]. Trimmed fragments from HISTA2 were counted using Stringtie (version:1.3.0) [14], normalized with trimmed mean of M (TMM) values [15], and trimmed with a Perl script to calculate the FPKM value for each gene. The EdgeR package (http://www.rproject.org/) was used for DEGs between samples, and a *p*-value was obtained for multiple hypothesis test correction [15]. Genes with |log2(Fold Change)|≥ 1 and an adjusted *p*-value < 0.01 were considered as DEGs. The *p*-values were adjusted using the Benjamini–Hochberg Procedure for controlling the false discovery rate [16].

### 2.3. Time Series Expression Pattern Clustering

To explore gene expression changes in different growth stages of WH pigs, STEM (Short Time-series Expression Miner) software [17] was used to cluster the expression levels of DEGs with similar expression patterns, and their expression patterns were analyzed. In this study, the method assumed that the logarithmic ratio of expression levels at each period relative to 60 days of age represented the gene expression value. Genes with similar expression trends could be grouped together to form a set of profiles determined by correlation coefficients. The method then selected a predetermined set of temporal model profiles and determined the statistical significance of the number of genes assigned to each profile compared to the number of genes expected based on opportunity. The value *p* < 0.05 was considered significant. The trend of significant enrichment would be color-coded in the final output trend diagram. The less obvious trends had no color.

### 2.4. Functional Annotation

Gene Ontology (GO) enrichment analysis of DEGs was conducted through the GO database to clarify the main biological functions of DEGs. The pathway enrichment analysis of DEGs was conducted through the KEGG database to elucidate the major signal transduction and biochemical metabolism pathways involved in DEGs.

### 2.5. Reliability Verification of RNA-seq Data

To verify that the results of DEGs screened by transcriptome sequencing data were legitimate, 12 DEGs were screened by RT-qPCR (Table 1). Using β-actin as the internal reference gene, the primers were designed using Primer Premier 6.0 software, and the primer sequences are shown in Table 1. The primers were synthesized by Sangon Bioengineering Co., Ltd. (Shanghai, China).

The total RNA was extracted from the LD muscle of WH pigs of different ages (60, 120, and 240 d) using Trizol reagent, and the RNA was reverse transcribed into cDNA using the Primescript RT reagent Kit (Thermo Fisher, Shanghai, China) and detected by gel electrophoresis. Then, qPCR was performed using the SYBR Premix Ex Taq^TM^ⅡKit (Thermo Fisher, Shanghai, China). The reaction system (20 μL) consisted of 10 μL SYBR Green qPCR Mix, 0.5 μL upstream/downstream primers (10 μmol/L), 1 μL cDNA, and 8 μL ddH_2_O. The PCR reaction conditions were as follows: pre-denaturation at 95 °C for 2 min, 40 cycles of denaturation at 95 °C for 15 s, annealing at 60 °C for 30 s, and extension at 72 °C for 30 s. For the melting curve collection, three biological replicates were taken per sample, and the relative gene expression levels were calculated by the 2^−ΔΔCt^ method; the SD of the threshold (Ct) value did not exceed 0.5. The experimental data were expressed as the mean ± standard deviation, and the difference was analyzed by a *t* test, with *p* < 0.05 as the criterion for the significance of difference.

### 2.6. Data Availability

All the original transcriptome sequencing data of LD muscle at different growth stages of WH pigs were uploaded to the NCBI database. The database entry number obtained is PRJNA915318.

## 3. Results

### 3.1. Body Weight and IMF Content

The IMF content and live weight for the WH pigs are shown in Figure 1. From 60–240 d, the IMF content of LD muscle increased significantly (*p* < 0.05). The IMF content at 240 d was significantly higher than that at 120 and 60 d (*p* < 0.05). The IMF content at 120 d was higher than that at 60 d, although not statistically significant (*p* > 0.05).

### 3.2. Summary of RNA-seq Results

Table 2 summarizes the RNA-seq results. The unique ratio of mapping to the reference genome (sus scrofa, version 11.1) ranged from 95.84–96.89%, with good alignment. Pearson’s Correlation Coefficient (R^2^) was used as an evaluation index for biological replicate correlation. The R^2^ for the sample expression were 95.0–96.0% for LD60-1, LD60-2, and LD60-3, 94.0–96.0% for LD120-1, LD120-2, and LD120-3, and 94.0–95.0% for LD240-1, LD240-2, and LD240-3. The results show that the experimental results were reliable, the experimental repeatability was good, and the selection of experimental samples was reasonable (Figure 2a).

The complexity of the data could be reduced by principal component analysis, and the relationships and variation in sizes among the samples at different developmental stages of the WH pigs could be deeply explored (Figure 2b). The contribution rate of the first principal components (PC1) was 30.8%, and that of the second principal component (PC2) was 22.5%. The aggregation of LD muscle samples at the same developmental stage indicated a high biological repeatability between the experimental samples.

### 3.3. Differential Expression Analysis

In this study, EdgeR [18] software was used to analyze the DEGs among samples, and the DEGs among different ages of WH pigs were screened (*p*-value < 0.01; |log_2_FC| ≥ 1). In total, 616, 485, and 1487 DEGs were identified for comparisons between LD60 and LD120, LD120 and LD240, and LD60 and LD240, respectively. Of these DEGs, 398, 255, and 1047 were up-regulated, and 218, 230, and 440 were down-regulated (Figure 3a–d). The co-expressed genes between the groups are shown in a Venn diagram, with 25 genes differentially expressed at different growth stages of WH pig (Figure 3e). The number of DEGs in the LD120 vs. LD240 group was lower than that in the LD60 vs. LD120 and LD60 vs. LD240 groups, and most genes were up-regulated in LD120 and LD240, which was consistent with the changes in muscle fiber area and IMF content [19].

For all the identified expressed genes, hierarchical clustering was carried out for the relationship between the samples and genes according to their gene expression levels, and the clustering results were presented using heatmaps (Figure 4). Each column represents a sample and each row represents a gene. Clustering analysis was used to explore the similarity of gene expression patterns in different samples.

### 3.4. STEM Analysis

To explore the dynamic changes of DEGs at different developmental stages of LD muscle in WH pigs, the STEM software (version 1.3.11) was used to perform a cluster analysis of the expression patterns of the 1899 DEGs screened out (Figure 5). A total of 15 DEG patterns were obtained, five of which were significantly enriched during muscle development (*p* < 0.05). Profiles 0, 2, 3, and 7 showed similar expression patterns, and the DEGs showed a significantly down-regulated trend during the growth and development of the LD muscle of WH pigs, while Profile 13 showed a significantly up-regulated trend. The continuously up-regulated/down-regulated DEGs related to fat metabolism and IMF deposition can be seen in Table 3.

### 3.5. Functional Annotation and Enrichment Analysis of DEGs

GO functional annotation was carried out to enrich and analyze the DEGs in the library of WH pigs at different developmental stages. Figure 6 shows the top 20 most significantly enriched GO items in each group. For LD60 vs. LD120, 489 GO items were significantly enriched (*p* < 0.05) and included “transition between fast and slow fiber”, “cardiac myofibril assembly”, “sarcomere organization”, and “regulation of the force of heart contraction” (Figure 6a). For LD60 vs. LD240, 1111 GO items were significantly enriched (*p* < 0.05), and included “response to interferon-α”, “MHC class II protein complex”, “fibrillar collagen trimer”, and “ISG15-protein conjugation” (Figure 6b). At LD120 vs. LD240, 676 GO items were significantly enriched (*p* < 0.05), and included “collagen trimer”, “extracellular matrix structural constituent”, “fibrillar collagen trimer”, and “p38MAPK cascade” (Figure 6c).

KEGG analysis of the DEGs determined significantly enriched pathways for the DEGs at different developmental stages, as shown in Figure 7. At LD60 vs. LD120, the DEGs were significantly enriched to 22 signaling pathways (*p* < 0.05) and included the “RIG-I-like”, “PPAR”, and “NOD-like receptor” signaling pathways (Figure 7a). At LD60 vs. LD240, the DEGs were significantly enriched to 94 signaling pathways (*p* < 0.05) and included the “B cell receptor”, “Chemokine”, and “NF-kappa B” signaling pathways (Figure 7b). At LD120 vs. LD240, the DEGs were significantly enriched to 30 signaling pathways (*p* < 0.05), and included the “FoxO”, “Rap1”, and “PI3K-Akt” signaling pathways, and “Fatty acid degradation” (Figure 7c). The results indicated that the period from 120–240 d was the key period of fat deposition in the WH pigs, which was consistent with the trend in the IMF.

### 3.6. Quantitative Validation of RNA-seq Data

Twelve DEGs related to growth and development or IMF, including *ACTC*, *CIART*, *HOXD1*, *NREP*, *PCK2*, *PHGDH*, *POSTH*, *S100A1*, *SPEGNB*, *TBKBP1*, *TP531NP2,* and *UBE2L6*, were randomly selected and verified by RT-qPCR. The results showed that the expression trends of the DEGs were consistent with the RT-qPCR results, and the results obtained by the two quantitative methods had a high correlation (R^2^ = 0.887), demonstrating that the results of the RNA-seq data were accurate and reproducible (Figure 8).

## 4. Discussion

IMF is a key factor affecting pork quality, and the higher meat quality of locally produced Chinese pigs over imported pigs is largely because of the higher IMF content. Therefore, increasing the IMF content is an effective way to improve the meat quality. In this study, we sought to identify IMF deposition-related genes by employing comparative transcriptome analyses on the LD muscle of WH pigs at three developmental stages (LD60, LD120, and LD240). These results improve our understanding of the molecular mechanisms of IMF deposition during skeletal muscle development, with the potential to accelerate the molecular breeding of pig meat traits and promote pig selection through modern molecular breeding techniques.

The cluster analysis of DEG expression patterns at three stages during LD muscle development in WH pigs using STEM software [20] found that five expression patterns (profiles 0, 2, 3, 7, and 13) were significantly enriched during the LD muscle development (*p* < 0.05). The expression patterns of profiles 0, 2, 3, and 7 were down-regulated from 60–240 d, while the expression patterns of profile 13 were up-regulated. The four significant expression genes with known functions related to muscle development and lipid metabolism were screened, including *S100A1* [21], *NR4A1* [22], *TGFBI* [23], and *CXCL10* [24]. The muscle fiber type is closely related to pork quality, which affects the growth and development of skeletal muscle, physical and chemical properties of the meat quality, and sensory evaluation. The number of muscle fibers in pigs is basically unchanged after birth, while muscle growth mainly depends on the hypertrophy of the muscle cells. The diameter of the muscle fibers increases with age and finally reaches a plateau [25]. Around the time of birth, pigs perform little exercise, the body has a high oxygen content, and most of the muscle fibers are oxidized. The amount of vigorous exercise increases with age, leading to an insufficient oxygen supply to the body, and the proportion of oxidized muscle fibers gradually decreases, while the proportion of glycolytic muscle fibers gradually increases [26]. Furthermore, the myoglobin content of glycolytic muscle fibers is considerably lower than that of oxidized muscle fibers. Therefore, with increasing age and numbers of glycolytic muscle fibers, insufficient oxygen supply or hypoxia can limit the hypertrophy process in WH pigs [27].

Known functional genes related to fat deposition, such as *SCD*, are mainly involved in the synthesis of unsaturated fatty acids [28]. The expression level of *SCD* in hybrid pigs of the breeds Large White and Landrace pigs was positively correlated with the monounsaturated fatty acid and total fatty acid content in the muscle [29]. *AGPAT2* is mainly involved in triglyceride synthesis and storage in adipocytes [30]. The *HSPB8* gene is mainly involved in the regulation of IMF, and the expression of this gene increases with age [31,32]. *LEPR* and *IRX3* genes are mainly involved in fat metabolism [33]. *MYL9* can promote endothelial cytoskeleton remodeling, stress fiber formation, myosin contraction, muscle development, and cell proliferation [34]. The expression of *MYL9* was significantly down-regulated with age [35]. *DLK1* encodes a glycosylated transmembrane protein, which plays an important role in inhibiting adipocyte differentiation and muscle growth [36]. *DLK1* sites in pigs increase the muscle mass and reduce fat deposition significantly, which is an important candidate gene for regulating the muscle development [37]. In this study, the expression levels of genes in LD60 were significantly lower than those in the LD240. The expression levels of genes in LD60 were significantly higher than those in LD240, which was consistent with the trend in the muscle fiber type transformation and IMF deposition during muscle development, indicating that the IMF deposition and enhanced muscle development phenotype were the result of the synergistic expression of multiple genes. In addition, Wang et al. found that the IMF of LD muscle at 240 days of age was significantly higher than that at 60 and 120 days of age, which was consistent with the results of this study [38].

The analysis of the signaling pathways can further help us to understand the biological functions of DEGs. In Figure 3e, 25 DEGs were found among the three comparison groups, and 7 KEGG pathways were found to be significantly enriched through the enrichment analysis (*p* < 0.05). These significantly enriched pathways were mainly related to muscle development, fat deposition, and energy metabolism. Myoblast differentiation is mediated by the HIF-1 signaling pathway [39], and the KEGG cluster analysis showed that the HIF-1 signaling pathway was the most significant (*p* < 0.05), indicating that the HIF-1 signaling pathway plays a key role in the three stages of muscle development. As myogenic progenitors, myoblasts in skeletal muscle can self-renew and generate new muscle fibers, and oxygen levels are critical for regulating the myogenesis capacity of myoblasts. Hypoxia inducible factors (HIFs) are key transcription factors that regulate oxygen homeostasis in tissue cells and accurately maintain the balance between the muscle growth and regeneration. Their expression is regulated by oxygen concentration, and they play a role in regulating cell energy metabolism, cell proliferation, and differentiation [40,41]. Genes activated by HIF-1 can enhance the oxygen delivery to tissues and promote cell metabolism to adapt to low oxygen levels, and HIF-1 negatively regulates the transcriptional regulation of various glycolytic pathway enzymes [42]. HIF-1 overexpression in hypoxic environments reduces the expression level of the myodifferentiation factor and myoietin, thus inhibiting myoblast differentiation [43]. Li et al. [44] found that HIF-1α is important in the transition from oxidative metabolism to glycolysis in muscles. In this study, *PFKFB3* was significantly enriched in the HIF-1 signaling pathway. The *PFKFB3* gene regulates the synthesis and degradation of fructose-2, 6-diphosphate, and fructose-6-phosphate during glucose metabolism [45]. *PFKFB3* plays an important role in muscle growth and development by regulating glucose metabolism [46]. The expression level of *PFKFB3* in LD60 was significantly higher than that in LD240, which may be related to the small amount of physical activity and high oxygen content in the body of WH pigs at birth. With increasing age, the amount of vigorous physical activity increases and results in insufficient oxygen supplies. Glycolysis is the main pathway involved in muscle development, thus inhibiting muscle cell differentiation and affecting the skeletal muscle development. Therefore, the DEGs in this study corresponding to significant enrichment pathways may play a key role in the postnatal growth and development and IMF deposition of WH pigs. These results indicate that the IMF deposition is the result of the synergistic development of multiple genes and pathways.

## 5. Conclusions

In this study, we performed a comparative analysis of the LD muscle transcriptome during three developmental stages in WH pigs. Many DEGs related to lipid metabolism and muscle growth development were identified, which may be involved in the regulation of the IMF deposition in WH pigs. Significant functional enrichment analysis of DEGs indicated that the biological pathway was directly related to the temporal changes in the skeletal muscle development of WH pigs. These candidate genes are significantly enriched in the key pathways involved in IMF deposition, such as the HIF-1 signaling, AMPK signaling, and fructose and mannose metabolism pathways. The results of this study contribute to our understanding of the molecular mechanism underlying IMF deposition and provide a new way to accelerate the genetic improvement of pork quality.

## Figures and Tables

**Figure 1 genes-14-00903-f001:**
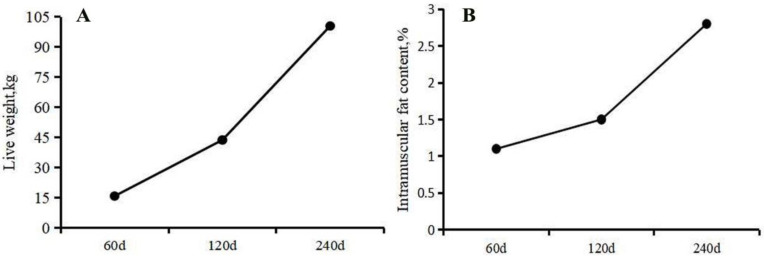
Phenotypic characteristics in the LD muscle of WH pigs. (**A**) Live weight. (**B**) IMF content of LD muscle over the three development stages.

**Figure 2 genes-14-00903-f002:**
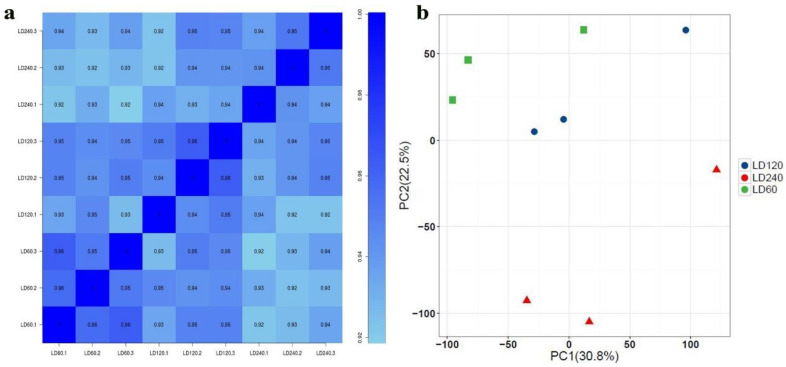
Cluster analysis and PCA. (**a**) Heatmap of correlation coefficients between LD muscles at different developmental stages of WH pig. (**b**) Principal components analysis (PCA).

**Figure 3 genes-14-00903-f003:**
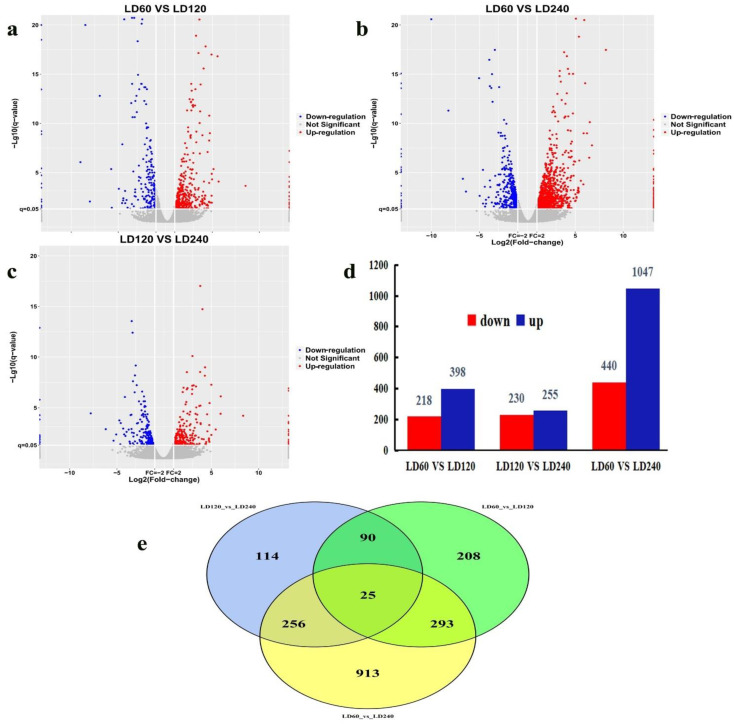
Volcano map and number of DEGs in different comparison groups at different developmental stages of WH pig. (**a**–**c**) Volcanic maps of DEGs in different comparison groups at different developmental stages. (**d**) Numbers of DEGs at different stages of WH pig. (**e**) Venn diagram of DEGs at different developmental stages of WH pig.

**Figure 4 genes-14-00903-f004:**
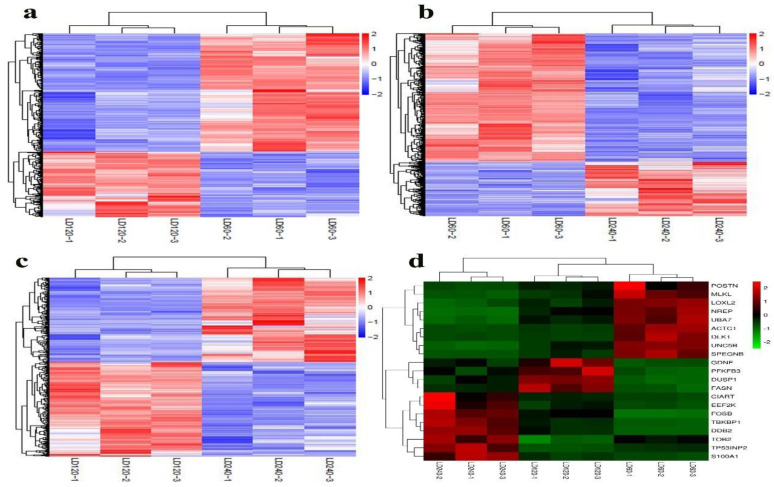
Hierarchical cluster analysis of 9 samples from three developmental stages of WH pig and expression heatmaps of DEGs at different developmental stages. (**a**–**c**) LD60 vs. LD120, LD120 vs. LD240, LD60 vs. LD240. (**d**) Heatmaps of IMF adipose-related DEGs at different developmental stages.

**Figure 5 genes-14-00903-f005:**
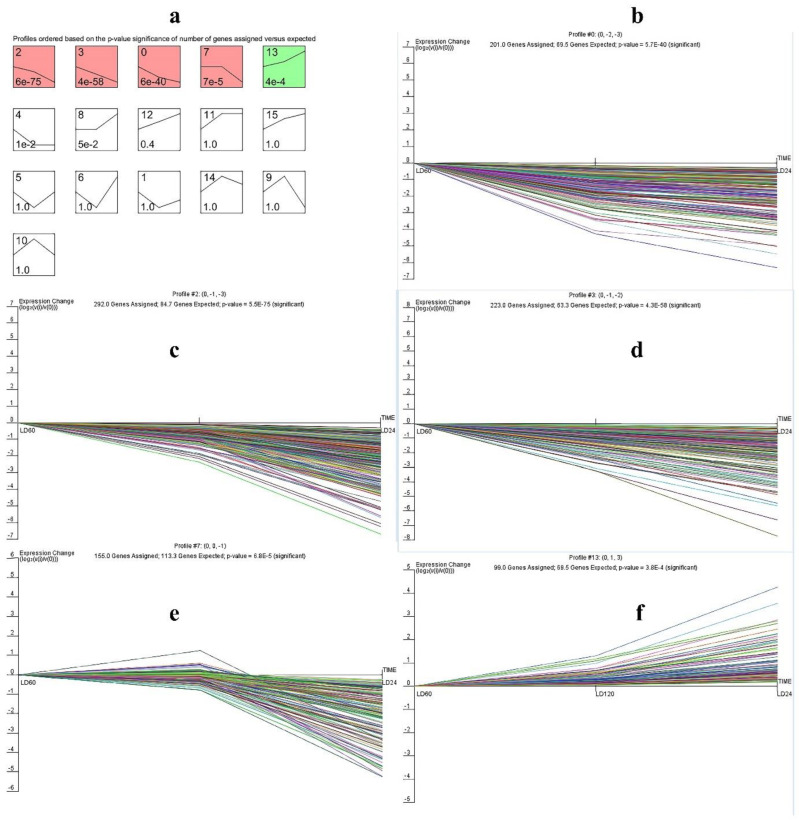
STEM cluster analysis of DEGs at different developmental stages in WH pigs. (**a**) A total of sixteen DEG patterns were obtained at different developmental stages of WH pig. (**b**–**f**) Five expression patterns of DEGs were significantly enriched in WH pigs.

**Figure 6 genes-14-00903-f006:**
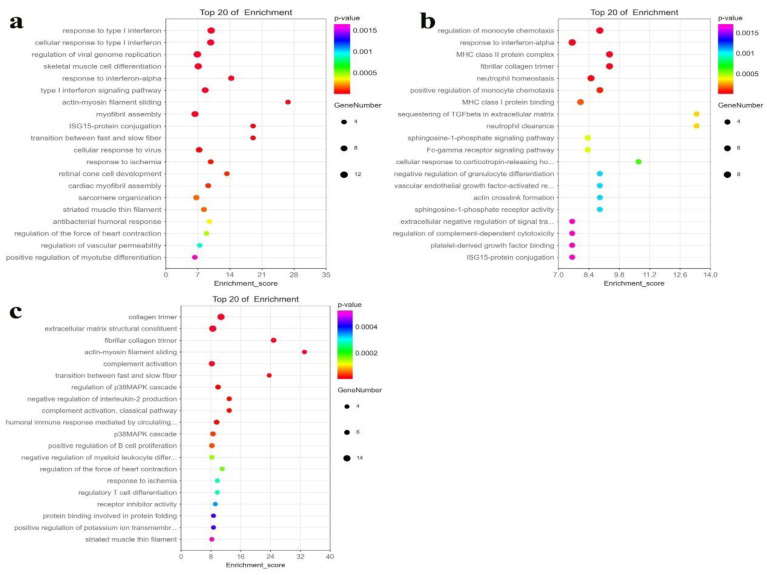
Significant GO terms of DEGs. (**a**) LD60 vs. LD120. (**b**) LD60 vs. LD240. (**c**) LD120 vs. LD240. Note: The *Y*-axis represents the GO term, and the *X*-axis represents the enrichment score. The larger the enrichment score, the greater the degree of enrichment. The size of the dots indicates the number of genes in the GO term. The color of the dots corresponds to different *p*-value ranges.

**Figure 7 genes-14-00903-f007:**
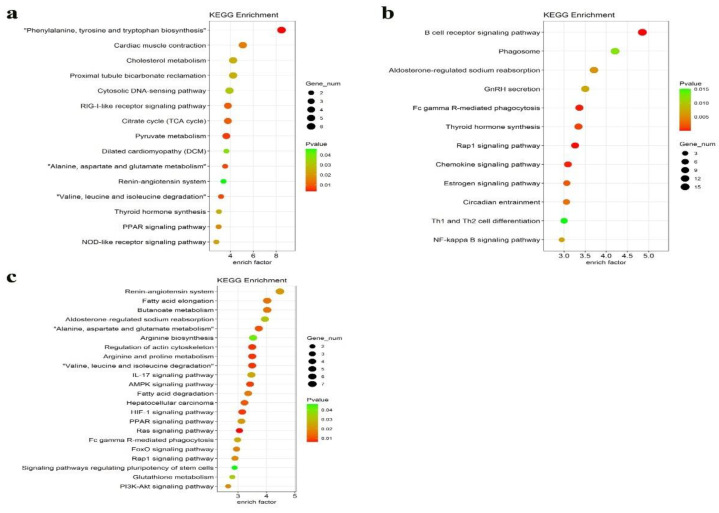
Significant KEGG terms of DEGs. (**a**) LD60 vs. LD120. (**b**) LD60 vs. LD240. (**c**) LD120 vs. LD240. Note: The *Y*-axis represents the name of the pathway, and the *X*-axis represents the enrich factor. The larger the enrich factor, the greater the degree of enrichment. The size of the dots indicates the number of genes in this pathway. The color of the dots corresponds to different *p*-value ranges.

**Figure 8 genes-14-00903-f008:**
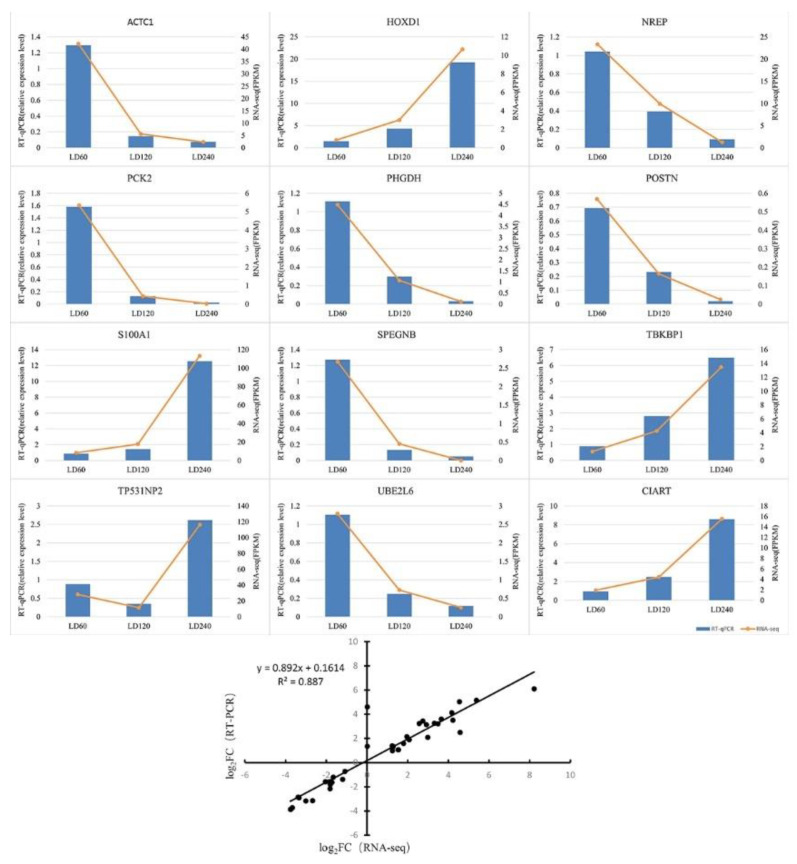
Experimental verification of gene expression level across three time points using RT-qPCR (*n* = 3). Note: The *Y*-axis on the left side of the histogram represents the gene expression level according to RT-qPCR, and the right *Y*-axis represents the standard value of FPKM based on transcription.

**Table 1 genes-14-00903-t001:** Primers used in qPCR for twelve genes.

Genes	Sequence (5′-to-3′)	Product Size (bp)	TM (°C)
*PCK2*	F: CCAACCAGAGGGCATCCAC R: AGTTACAATCACCGTCTTGCT	175	59.4
*ACTC1*	F: TCTGCTGGCATCCATGAAACGA R: GGCAGTGATTTCCTTTTGCAT	147	59.2
*PHGDH*	F: GAAATCGCCATCCAGTTCGT R: AGAGCTTCAGCCAGACCAA	125	56.3
*POSTN*	F: TCCCCATAACTGTCTACACC R: CCTTCCGTTTTGATAATAGGC	100	53.4
*HOXD1*	F: AGATAGCCAACTCATTGCAG R: GCAAAAGCCCTTCTCGTTC	99	54.5
*TBKBP1*	F: AACGGCTCAACCAATTCCA R: TCACTCTTCTCCACGCAGA	106	55.6
*TP53INP2*	F: CTGGCTCATCATTGACCTG R: TGCTGCCAGTAACATAAACTGA	119	53.7
*UBE2L6*	F: TGCCCATCATCAGCAACGA R: GTCTATTCACCAGCACGTTGA	84	52.6
*S100A1*	F: GAACCTGCTCCCAATACACC R: GAGTCTCCATCGCTGTCTC	76	56.0
*SPEGNB*	F: CATCAACATAACCAACCCCTT R: TTAATCTTCGCCGGGACTTCCA	75	58.5
*NREP*	F: CCAAAGGAAGTGAACCGCAAG R: TCATTTTCATTGCTGCCGAGT	77	59.0
*CIART*	F: GGAGCCAAGTCTAAATATCCAC R: TCCTCTTTCAAATCGACCCAT	139	56.8
*β-actin* ^a^	F: GTCATCACCATCGGCAATGAG R: AATGCCGCAGGATTCCATG	84	56.4

^a^ *β-actin* is the internal control gene.

**Table 2 genes-14-00903-t002:** Summary statistics of the transcriptome.

Samples ID	Raw Reads	Clean Reads	Mapped Reads	Clean Ratio	Unique Reads	Mapping Ratio
LD60-1	50,054,498	45,849,605	42,757,962	91.60%	42,502,777	96.28%
LD60-2	54,562,360	50,025,837	46,411,138	91.69%	46,130,037	96.24%
LD60-3	51,280,966	46,622,900	43,592,197	90.92%	43,333,563	96.46%
LD120-1	50,000,566	46,569,727	43,889,677	93.14%	43,674,962	96.89%
LD120-2	53,709,810	48,261,833	44,860,139	89.86%	44,592,694	95.87%
LD120-3	53,573,860	49,887,575	46,710,172	93.12%	46,427,981	96.05%
LD240-1	48,876,746	43,937,385	41,037,844	89.89%	40,835,618	96.72%
LD240-2	53,312,294	49,045,957	45,672,971	92.00%	45,385,253	96.02%
LD240-3	53,267,442	48,786,293	45,298,674	91.59%	45,048,382	95.84%

**Table 3 genes-14-00903-t003:** Differentially expressed genes associated with IMF deposition at different developmental stages of WH pigs.

Gene Name	Description	LD60 vs. LD120		LD60 vs. LD240		LD120 vs. LD240	
		log2FC	*p*-Value	log2FC	*p*-Value	log2FC	*p*-Value
*ACTC1*	actin α cardiac muscle 1	2.92	9.29E-16	4.17	5.69E-29	1.25	9.40E-04
*TGFBI*	transforming growth factor β induced	0.21	8.74E-01	1.28	1.26E-04	1.08	1.73E-03
*UCP2*	uncoupling protein 2	0.25	9.92E-01	2.16	1.22E-04	1.92	8.37E-04
*TNNT2*	Sus scrofa troponin T2, cardiac type (TNNT2).	1.63	7.00E-03	3.77	2.89E-07	2.14	1.32E-02
*FAM184B*	family with sequence similarity 184 member B	0.59	2.29E-01	1.45	7.47E-04	0.85	3.16E-02
*FADS1*	fatty acid desaturase 1	0.47	4.59E-01	1.59	2.46E-03	1.11	2.52E-02
*WWTR1*	WW domain containing transcription regulator 1	0.58	2.56E-01	1.19	2.52E-03	0.61	7.04E-02
*MYL9*	myosin light chain 9	1.24	2.01E-03	1.59	4.38E-03	0.34	5.50E-01
*SH3PXD2B*	SH3 and PX domains 2B	0.63	2.94E-01	1.63	3.14E-03	1.00	5.45E-02
*NCF1*	neutrophil cytosolic factor 1	0.40	8.08E-01	4.13	2.58E-04	3.73	8.47E-04
*ANXA1*	annexin A1	1.14	2.37E-02	1.72	6.31E-04	0.58	1.72E-01
*FGFR4*	fibroblast growth factor receptor 4	−0.10	3.00E-01	1.52	3.62E-04	1.62	7.17E-05
*PHGDH*	phosphoglycerate dehydrogenase	2.07	2.08E-05	5.38	1.72E-22	3.31	1.32E-07
*POSTN*	periostin	1.79	1.69E-03	4.54	5.81E-11	2.75	1.24E-05
*TP53INP2*	tumor protein p53 inducible nuclear protein 2	1.30	2.58E-05	−2.05	2.22E-09	−3.35	1.49E-16
*NREP*	neuronal regeneration related protein	1.24	4.18E-05	4.22	5.26E-27	2.99	3.53E-14
*DLK1*	delta like non-canonical Notch ligand 1	4.07	3.64E-19	7.52	1.12E-27	3.45	1.06E-03
*HSPB8*	heat shock protein family B (small) member 8	−0.32	5.09E-02	−1.06	3.45E-04	−0.74	4.32E-02
*SETD7*	SET domain containing 7, histone lysine methyltransferase	−0.49	3.70E-02	−1.39	1.39E-05	−0.91	3.13E-02
*S100A1*	S100 calcium binding protein A1	−1.09	9.64E-04	−3.76	5.85E-17	−2.67	5.19E-09
*SCD*	stearoyl-CoA desaturase	−0.37	2.52E-01	−2.21	1.61E-03	−1.84	7.84E-03
*LEPR*	leptin receptor	−1.30	3.96E-01	−4.32	2.94E-04	−3.03	1.21E-02
*IRX3*	iroquois homeobox 3	−0.69	3.22E-02	−2.41	1.21E-05	−1.72	3.19E-03
*AGPAT2*	1-acylglycerol-3-phosphate O-acyltransferase 2	−0.33	5.44E-02	−1.04	6.77E-04	−0.71	9.00E-02

## Data Availability

Not applicable.

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
