# Peer review of "Transcriptome Profiling of Different Developmental Stages on Longissimus Dorsi to Identify Genes Underlying Intramuscular Fat Content in Wannanhua Pigs"

_genes, 2023, doi:10.3390/genes14040903_

Round 1
Reviewer 1 Report
The overall transcriptome data analysis of the study needs to be improved. The only experimental data that supports the RNA-Seq data are the qPCR data. If possible, this this should be supplemented with how the histology of LD changes during the three development stages. This can be easily achieved via H&E staining of LD sections.
Since the entire study depends on RNA-Seq data analysis, the bioinformatic analysis of the RNA-Seq data needs to be improved. This can be done via methods such as weighted gene co-expression network analysis (WGCNA) that could potentially enable the idenitification of gene regulatory networks that may play an important role in IMF deposition.
More specific comments with regard to data analysis and english language can be found in the attached manuscript as inline comments. Each highlighted section carries a request for clarifications or suggestions for improvements.

Author Response
1.The overall transcriptome data analysis of the study needs to be improved. The only experimental data that supports the RNA-Seq data are the qPCR data. If possible, this this should be supplemented with how the histology of LD changes during the three development stages. This can be easily achieved via H&E staining of LD sections.
Reply: Thank you very much for your suggestion. We have studied the histological changes of longissimus dorsi muscle in different developmental stages of Wannan flower pigs. Related articles have been published in (Jia Zhang. 2010. Effects of adiponection on skeletal muscle growth and types of muscle fibers of Wannanhua pigs. Anhui Agricultural University (in Chinese).) Pigs. Anhui Agricultural University (in Chinese).) Therefore, no relevant research was done this time.
2.Since the entire study depends on RNA-Seq data analysis, the bioinformatic analysis of the RNA-Seq data needs to be improved. This can be done via methods such as weighted gene co-expression network analysis (WGCNA) that could potentially enable the idenitification of gene regulatory networks that may play an important role in IMF deposition.
Reply: Thank you for your opinion. WGCNA analysis was helpful to construct molecular regulatory networks related to IMF content in different developmental stages of Wannan flower pigs. However, since the number of samples in this study was 9 samples in 3 developmental stages, and the minimum number of samples for WGCNA analysis required 15 samples, WGCNA analysis was not conducted. In the later stage, we must consider increasing the number of samples, and then carry out WGCNA analysis.
3.More specific comments with regard to data analysis and english language can be found in the attached manuscript as inline comments. Each highlighted section carries a request for clarifications or suggestions for improvements.
Reply: Thank you very much for your suggestion. It has been revised in the newly submitted paper.
Reviewer 2 Report
The study is aimed at the in depth analysis of gene expression and genetic regulation during intramuscular fat development in Wannanhua pigs using RNA-seq data. The study is well-designed and clearly described. The language needs only subtle corrections. Nevertheless, the manuscript has some flaws that need to be addressed:
Methods section:
- Please complete the description of the methods with the missing names of the reagents used for cDNA libraries preparation, reverse transcription, etc.
- Line 127: The qPCR results require statistical analysis, please include it in the manuscript
Results section:
-Line 160: Please add to the supplementary materials a table containing the identified DEGs, log2FC, and p-values corresponding to genes
-Line 187: Please add the number of DGEs in the 5 statistically significant clusters
-Line 201: Please add to the supplementary materials a table containing the identified GO and KEGG significantly enriched pathways
-Line 230: as above, were the obtained PCR results statistically significant?
Discussion section:
- I would suggest that the authors briefly discuss the obtained results with the available data for transcriptome analysis in intramuscular fat in other pig breeds.
Author Response
The study is aimed at the in depth analysis of gene expression and genetic regulation during intramuscular fat development in Wannanhua pigs using RNA-seq data. The study is well-designed and clearly described. The language needs only subtle corrections. Nevertheless, the manuscript has some flaws that need to be addressed:
Methods section:
- Please complete the description of the methods with the missing names of the reagents used for cDNA libraries preparation, reverse transcription, etc.
Reply: Thank you very much for your suggestion. It has been changed in the article.
- Line 127: The qPCR results require statistical analysis, please include it in the manuscript
Reply:Thank you very much for your suggestion. It has been changed in the article.
Results section:
-Line 160: Please add to the supplementary materials a table containing the identified DEGs, log2FC, and p-values corresponding to genes
Reply: Thank you very much for your suggestion. It has been changed in the article.
-Line 187: Please add the number of DGEs in the 5 statistically significant clusters
Reply: Thank you, the number of differential genes has been marked in Figure 5, sorry, maybe because the pixel in Figure 5 is not very high, so it is not very clear, now it has been corrected.
-Line 201: Please add to the supplementary materials a table containing the identified GO and KEGG significantly enriched pathways
Reply: Thank you very much for your suggestion. Added to attached material.
-Line 230: as above, were the obtained PCR results statistically significant?
Reply: Thank you very much for your suggestion. It makes sense. Correlation analysis results of transcriptome data and fluorescence quantitative PCR data have been added, and correlation coefficient R2=0.887, indicating that transcriptome data is real and reliable.
Discussion section:
- I would suggest that the authors briefly discuss the obtained results with the available data for transcriptome analysis in intramuscular fat in other pig breeds.
Reply: Thank you very much for your suggestion. It has been changed in the article.
Reviewer 3 Report
I am confused as to what was the objective of this study. If it was to understand transcriptome involved in IMF deposition, then this study design is invalid (more on this below). More over the title is wrong, this study has not done any spatial transcriptome analysis, you have only sampled at different developmental stages of pig, the title must be corrected to reflect that. Moreover your transcriptome study design only allows you to understand transcriptional difference between development stages.
Since this is not an animal science / swine science journal, it will be better discuss in the introduction about the choice of sampling time points and also provide more information on WH pigs and why IMF is considered important and their role in meat quality.
What was the RIN threshold used for selecting samples ?
How were low quality sequence identified and removed, make the perl script available, without that its not possible to verify if the right thresholds were applied.
Hisat must be HISAT thats the name of the software.
Double check the accession number given, at this time it does not show up in SRA website.
Line 69 - 72 How many individuals did you use ? You say 6 castrated males were used, then say n = 3 and then in the next sentence you say 3 individuals per stage (meaning totally 9) its not clear how many samples were used.
Line 186 how did you arrive at 1,899 DEGs ? In line 163 616, 485, and 1,487 degs were found that would mean, cumulatively there are ~ 2400 DEGs
Fig 5 quality is very poor. Higher resolution figure must be supplied, nothing is readable at this stage.
Table 3 :- How were these representative IMF deposition genes arrived at ? Did you find Fatty acid metabolism related genes from your DEG list ?
What type of clustering was used for STEM analysis
Lines 244 - 245 "These results improve our understanding of the molecular 244
mechanisms of IMF deposition during skeletal muscle development"
I do not agree with this statement. Since you have compared LD muscle at different stages of pig development
'most of the difference in gene expression that you have found can be attributed to developmental stages rather than difference in IMF content.
If your intention was to identify genes involved in differences in IMF deposition , this study design does not allow you to find that. For that you will need
to identify animals differing in IMF content and perform transcriptome analysis.
Gene names must be in italic.
lines 268 - 284 Please point out which of these were DEGs in your study and how they related with LD development in WH.
Line 287 - What do you mean by 25 DEGs found among three comparsion ? do you mean to say 25 genes were DE in all three comparison
Line 311 - did you measure oxygen level ? If not, on what basis if it concluded that oxygen levels are high in WH ?
There a number of studies that have looked at transcriptome difference in LD of pigs at D60, D120 and D240, how does you results compare to them ?
Author Response
1.I am confused as to what was the objective of this study. If it was to understand transcriptome involved in IMF deposition, then this study design is invalid (more on this below). More over the title is wrong, this study has not done any spatial transcriptome analysis, you have only sampled at different developmental stages of pig, the title must be corrected to reflect that. Moreover your transcriptome study design only allows you to understand transcriptional difference between development stages.
Reply: Thank you for your comments. The title of the paper has been changed.
2.Since this is not an animal science / swine science journal, it will be better discuss in the introduction about the choice of sampling time points and also provide more information on WH pigs and why IMF is considered important and their role in meat quality.
Reply: Thank you very much for your suggestion. It has been changed in the article.
3.What was the RIN threshold used for selecting samples ?
Reply: Thank you very much for your suggestion. The minimum RIN threshold value of the sample used in this study was 8.2, and the minimum 28S/18S value was 1.5, which met the requirements of RIN≥7 and 28S/18S≥0.7 of sequencing.
4.How were low quality sequence identified and removed, make the perl script available, without that its not possible to verify if the right thresholds were applied.
Reply: Thank you very much for your suggestion. In this study, SEQTK software was used to filter the data, which mainly included removing the junction sequences contained in reads. Remove base with mass Q below 20 at the 3 'end, that is, base error rate is less than 0.01, where Q= -10logerror_ratio; reads less than 25 in length were removed. Remove ribosome RNA reads from the species in which it belongs.
5.Hisat must be HISAT thats the name of the software.
Reply: Thank you very much for your suggestion. It has been changed in the article.
6.Double check the accession number given, at this time it does not show up in SRA website.
Reply: Yes, it's classified for now because of a confidentiality agreement, and the data will be released in a year.
7.Line 69 - 72 How many individuals did you use ? You say 6 castrated males were used, then say n = 3 and then in the next sentence you say 3 individuals per stage (meaning totally 9) its not clear how many samples were used.
Reply: Thank you very much for your suggestion. It has been changed in the article.
8.Line 186 how did you arrive at 1,899 DEGs ? In line 163 616, 485, and 1,487 degs were found that would mean, cumulatively there are ~ 2400 DEGs
Reply: Thank you very much for your suggestion. In STEM cluster analysis of differential genes, genes with 0 gene expression at three developmental stages were excluded.
9.Fig 5 quality is very poor. Higher resolution figure must be supplied, nothing is readable at this stage.
Reply: Thank you very much for your suggestion. It has been changed in the article.
10.Table 3 :- How were these representative IMF deposition genes arrived at ? Did you find Fatty acid metabolism related genes from your DEG list ?
Reply: Thank you very much for your suggestion. The genes related to IMF deposition in Table 3 were all selected from the differential genes screened from the transcription data in this study, and these differential genes were selected for RT-qPCR experiment to verify the accuracy of the RNA-seq data.
11.What type of clustering was used for STEM analysis.
Reply: This study mainly studied three different developmental stages of longissi dorsi muscle of Anhui Nanhua pigs. Therefore, the study on the expression trend of differential genes at different developmental stages is helpful to understand the dynamic changes of IMF deposition in Anhui Nanhua pigs. Compared with the static gene expression model of the same size, the temporal gene expression model had more information of derivable gene regulatory network. Therefore, STEM software was used in this study to cluster the expression levels of differentially expressed genes with similar expression patterns and analyze their expression patterns.
12.Lines 244 - 245 "These results improve our understanding of the molecular mechanisms of IMF deposition during skeletal muscle development". I do not agree with this statement. Since you have compared LD muscle at different stages of pig development'most of the difference in gene expression that you have found can be attributed to developmental stages rather than difference in IMF content.If your intention was to identify genes involved in differences in IMF deposition , this study design does not allow you to find that. For that you will need to identify animals differing in IMF content and perform transcriptome analysis.
Reply:Thank you. The IMF content of Wannanhua pigs in different developmental stages is different, and there is statistical difference, which has been described in the results section of this paper. In this paper, transcriptome techniques were used to screen out differentially expressed genes that regulate IMF at different developmental stages, and cluster analysis of time series expression patterns of DEGs was used to study the main molecular mechanisms that lead to IMF differences in different developmental stages of Wannanhua pigs.
13.Gene names must be in italic.
Reply:Thank you. It has been changed in the article.
14.lines 268 - 284 Please point out which of these were DEGs in your study and how they related with LD development in WH.
Reply: Thank you. The genes listed in Lines 268-284 are all differential genes screened in this study, and these genes have been proved to be related to muscle growth and development and intramuscular fat deposition, thus proving that the differential genes screened in this study are consistent with the results of previous studies.
15.Line 287 - What do you mean by 25 DEGs found among three comparsion ? do you mean to say 25 genes were DE in all three comparison
Reply:During the screening of differential genes at three different developmental stages, it was found that 25 differential genes were expressed in the longissimus dorsi muscle of Wannan swine at three different time periods.
16.Line 311 - did you measure oxygen level ? If not, on what basis if it concluded that oxygen levels are high in WH ?
Reply: Thank you very much for your suggestion. No detection analysis was conducted. According to relevant literature studies, IMF content gradually increased with the aging of pigs, and the proportion of glycolic muscle fibers gradually increased, while the proportion of oxidized muscle fibers gradually decreased. The oxygen content of the colytic muscle fiber is low, and the oxygen content of the oxidized muscle fiber is high.
17.There a number of studies that have looked at transcriptome difference in LD of pigs at D60, D120 and D240, how does you results compare to them ?
Reply: Thank you very much for your suggestion. It has been added to the paper. The results of this study are consistent with those of other studies.
Reviewer 4 Report
This is an interesting manuscript examining muscle gene expression relative to IMF deposition over time. My biggest concern is that there is the bare minimum of biological replication in this study and as such it needs to be acknowledged in the text of the manuscript that interpretation of this study needs to be taken cautiously due to the very small sample size.
This is exacerbated in the examination of the pathway and function enrichment analysis. The sample size is really too low for this type of analysis and a minimum of 4-5 samples per treatment is really necessary to get meaning from this type of analysis.
Line 67-77- the experimental design here is unclear and should be explained more clearly. n of 3 with 3 time points but 6 pigs?
Line 74: need the elapsed time between the time of slaughter and sample freezing.
Is the same multiple comparison corrections used for DEG and enrichment analyses? Please clarify in the text.
Figure 5 and Figure 7 are impossible to read and are fuzzy when zooming in to read them.
Line 189-191- the size of the font changes here and should be made uniform
Author Response
1.This is an interesting manuscript examining muscle gene expression relative to IMF deposition over time. My biggest concern is that there is the bare minimum of biological replication in this study and as such it needs to be acknowledged in the text of the manuscript that interpretation of this study needs to be taken cautiously due to the very small sample size.
This is exacerbated in the examination of the pathway and function enrichment analysis. The sample size is really too low for this type of analysis and a minimum of 4-5 samples per treatment is really necessary to get meaning from this type of analysis.
Reply: Thank you for your comments.
2.Line 67-77- the experimental design here is unclear and should be explained more clearly. n of 3 with 3 time points but 6 pigs?
Reply: Thank you. It has been changed in the article.
3.Line 74: need the elapsed time between the time of slaughter and sample freezing. Is the same multiple comparison corrections used for DEG and enrichment analyses? Please clarify in the text.
Reply: Thank you very much for your suggestion. It has been changed in the article.
4.Figure 5 and Figure 7 are impossible to read and are fuzzy when zooming in to read them.
Reply: Thank you very much for your suggestion. It has been changed in the article.
5.Line 189-191- the size of the font changes here and should be made uniform
Reply: Thank you very much for your suggestion. It has been changed in the article.
Round 2
Reviewer 2 Report
Thank you for the changes made to the manuscript, it has vastly improved as a result of your hard work.
Minor issue:
- line 267 typo, not Duantitative but Quantitative
- as suggested, the authors added the statistical analysis of the qPCR technique in the methods but forgot to include the results of this analysis in the Results section. The correlation between the two used methods (qPCR vs RNA-seq) came out very nicely, but please provide information about the p-values you obtained for the 12 analyzed genes, and whether the results were statistically significant.
Author Response
- line 267 typo, not Duantitative but Quantitative
Reply: Thank you very much for your suggestion. It has been changed in the article.
- as suggested, the authors added the statistical analysis of the qPCR technique in the methods but forgot to include the results of this analysis in the Results section. The correlation between the two used methods (qPCR vs RNA-seq) came out very nicely, but please provide information about the p-values you obtained for the 12 analyzed genes, and whether the results were statistically significant.
Reply: Thank you very much for your suggestion. Correlation linear plots of qPCR and RNA-Seq data have been added in the paper.
Reviewer 3 Report
I have no further comments.
Author Response
Thank you for your review of the manuscript.
Reviewer 4 Report
The only remaining comment I have is that the samples size is still limiting and I dont think that is adequately addressed in the manuscript.
Author Response
The only remaining comment I have is that the samples size is still limiting and I dont think that is adequately addressed in the manuscript.
Reply: Thank you very much for your suggestion. The scientific research of pigs, cattle and sheep belongs to the research category of large animals, which needs to be considered from the overall design of the experiment, the expenditure of funds and the difficulty of sample collection. The current transcriptomic sequencing belongs to high-throughput sequencing with a depth of 10X and a large amount of data obtained. Therefore, from the perspective of reliability and rigor of experimental data results, at least 3 animals can be selected for each group when grouping omics studies of large animals. For example, the selection of the number of pig samples in each group in these two papers (Zhang D, Ma S, Wang L, et al. Min pig skeletal muscle response to cold stress. PLoS One. 2022, 17 (9) : e0274184. Doi: 10.1371 / journal pone. 0274184. or Yang Y, Fan X, Yan J, et al. A comprehensive epigenome atlas reveals DNA methylation regulating skeletal muscle development. Nucleic Acids Res. 2021, 49(3):1313-1329. doi: 10.1093/nar/ gkaa1203.) .